# Influence of Essential Oils on the Microbiological Quality of Fish Meat during Storage

**DOI:** 10.3390/ani11113145

**Published:** 2021-11-04

**Authors:** Simona Kunová, Esther Sendra, Peter Haščík, Nenad L. Vukovic, Milena Vukic, Miroslava Kačániová

**Affiliations:** 1Faculty of Biotechnology and Food Sciences, Institute of Food Sciences, Slovak University of Agriculture, Tr. A. Hlinku 2, 94976 Nitra, Slovakia; simona.kunova@uniag.sk (S.K.); peter.hascik@uniag.sk (P.H.); 2Centro de Investigación e Innovación Agroalimentaria y Agroambiental (CIAGRO-UMH), Escuela Politécnica Superior de Orihuela, Miguel Hernández University, 03312 Orihuela, Spain; esther.sendra@umh.es; 3Department of Chemistry, Faculty of Science, University of Kragujevac, R. Domanovica 12, 34000 Kragujevac, Serbia; nvukovic@kg.ac.rs (N.L.V.); milena.vukic@pmf.kg.ac.rs (M.V.); 4Faculty of Horticulture and Landscape Engineering, Institute of Horticulture, Slovak University of Agriculture, Tr. A. Hlinku 2, 94976 Nitra, Slovakia; 5Department of Bioenergetics, Food Analysis and Microbiology, Institute of Food Technology and Nutrition, Rzeszow University, Cwiklinskiej 1, 35-601 Rzeszow, Poland

**Keywords:** rainbow trout, *Cinnamomum camphora*, *Citrus limon*, total viable counts, coliform bacteria, lactic acid bacteria, mass spectrometry, MALDI-TOF

## Abstract

**Simple Summary:**

Fish meat is highly perishable due to its composition and the naturally present microbiota. The food industry aims to provide healthy, safe, and high-quality products to the market. Several strategies, including the use of natural preservatives, may be used to enhance food shelf life, and they can also be combined with others, such as vacuum packaging. This being the case, essential oils are natural plant components that, due to their composition, possess high antimicrobial and antioxidant effects, and are therefore good candidates to be tested as fish preservatives together with vacuum packaging. In the present study, essential oils from *Citrus lemon* and *Cinnamomum camphora* were added to rainbow trout meat for evaluating the microbiological quality (counts of bacteria and identification of present microbiota) of the fish when vacuum packed and stored for 7 days at 4 °C. Our results show that lemon (0.5% and 1%) as well as *C. camphora* essential oils (0.5% and 1%) had a positive effect on the microbiological quality of fish meat, keeping a high microbial quality of the fish fillets during 7 days of cold storage. The use of these essential oils in combination with vacuum packaging is beneficial in extending the shelf life of rainbow trout meat. All isolated species under the tested conditions are identified in the present study; such information will be useful for the future development of preservation methodologies that target isolated microorganisms, which will enable the food industry to enhance the shelf life and safety of fish.

**Abstract:**

The aim of the present study was to evaluate the microbiological quality of rainbow trout meat treated with essential oils (EOs from *Citrus limon* and *Cinnamomum camphora*) at concentrations of 0.5% and 1.0% in combination with vacuum packaging during storage. The composition of the EOs were analyzed by gas chromatography coupled with mass spectrometry, and total viable counts (TVCs), coliform bacteria (CB), and lactic acid bacteria (LAB) were determined on the zeroth, first, third, fifth, and seventh days of storage at 4 °C. Individual species of isolated microorganisms were identified using a MALDI-TOF MS Biotyper. The results show that the major components of the EOs were linalool (98.1%) in *C. camphora* and α-limonene in *C. limon*. The highest number of TVCs and CB were 4.49 log CFU/g and 2.65 log CFU/g in aerobically packed samples at the seventh day. The lowest TVCs were those of samples treated with 1% *C. camphora* EO. For CB the most effective treatment was 1% lemon EO. LAB were only detected in a few samples, and were never present in aerobically packed samples; the highest number of LAB was 1.39 log CFU/g in samples treated with 1% lemon EO at day seven. The most commonly isolated coliform bacteria were *Hafnia alvei*, *Serratia fonticola*, *Serratia proteamaculans*, *Pantoea agglomerans*, and *Yersinia ruckeri. Lactobacillus sakei*, *Staphylococcus hominis*, and *Carnobacterium maltaromaticum* were the most frequently isolated bacteria from lactic acid bacteria. In conclusion, *C. camphora* EO at a concentration of 1% showed the highest antimicrobial activity.

## 1. Introduction

Fish and fishery products are valued as nutritious and healthy foods. However, it is well-known that fish are highly perishable due to their high water activity, neutral pH, high content of low-molecular-weight molecules, and a microbiota adapted to low temperatures, which, altogether, are excellent conditions for biochemical and microbial spoilage. This is the reason why traditional methods for fish preservation (salting, drying, and freezing) involve a dramatic decrease in water activity. However, at present, consumers prefer fresh food that has been minimally processed. Nowadays, the estimates say that 20% of fish becomes spoiled post-catch [1]. Three different mechanisms have been defined for fish spoilage [2]: autolytic spoilage (mainly proteolysis, but also lipolysis), oxidative spoilage (oxidation of unsaturated fatty acids), and microbial spoilage (mainly the proliferation of psychrotolerant species with the production of biogenic amines). Fish waste is a major problem in a world with a growing population and limited resources, and, additionally, fish spoilage is a food safety issue and poses huge economical losses to food companies.

The use of natural preservatives is being investigated since many consumers perceive synthetic preservatives as having potential health risks [3]. Essential oils (EOs) are aromatic extracts (mainly terpenes and other aromatic compounds) from whole or specific parts of plants, whose bioactive components provide biological properties (antioxidant, insecticidal, antimicrobial, etc.). Such properties are a function of the plant itself, as well as many other factors (agronomic, part of the plant, extraction method, application method, etc.). EO solubility and bacterial type (Gram-negative bacteria are more resistant than Gram-positive bacteria) greatly influence antimicrobial properties; furthermore, the food matrix may interfere with EO activity [4]. EOs are valued for food preservation as they are natural products, many of them widely consumed by humans with a long history of being safe and its addition to foods is included in the present trend of ‘clean label’ foods.

Many authors have demonstrated the significant antimicrobial and antioxidant effects of a wide spectrum of EOs. Oregano, rosemary, and thyme EOs are the most commonly used natural antimicrobials to extend the shelf life of fish meat [5]. The effectiveness of EOs depends on the chemical structure, concentration, comparison of the spectrum of antimicrobial activity with the target microorganisms, interactions with the food matrix, and the method of application [6]. Regarding the food matrix, some authors have suggested that fat levels in fish may affect the effectiveness of essential oils, as some essential oils have been reported to be more effective in low-fat fish (cod) than in fatty fish (salmon) [7].

Several authors recently reviewed the use of EOs for fish preservation [1,2]. They evaluated the available scientific literature and pointed to origanum, thymus, and Zingiberaceae as the widest spectrum efficient EOs for the preservation of fish. They also pointed out the high relevance of the application method of EOs (need of hurdle technology, active film, and proper packaging) to achieve an extended shelf life, enhanced food safety, and sensory acceptability of fresh fish. Further investigations are needed to define clear guidelines which can be utilized to be able to effectively extend fish shelf life by using EOs, without impairing fish sensory properties.

The antimicrobial effects of different essential oils on freshwater fish as a strategy to reduce antimicrobials in aquaculture has been recently investigated by several authors [8]. In their study, they evaluated the composition of 14 essential oils and their inhibitory activity (disc diffusion assay, DDA, and minimum inhibitory concentration, MIC) against 20 bacteria isolated from freshwater fish. *Cinnamomum camphora* var. *Linaloolifera* EO was the most active EO against the majority of the isolates, especially against *Aeromonas* spp. and *Enterococcus*. The major component of the EO was linalool (>98%), showing high antibacterial activity in the study, whereas other authors have reported low antifungal activity of *C. camphora* L EO rich in linalool [9]. The EO of *Citrus lemon* was also tested in the study: inhibitory results were not as good; however, lemon EO has shown antibiofilm-forming properties in previous studies on food matrices [10]. Additionally, in aquaculture, the supplementation of the diets of rainbow trout with lemon EO [11], as well as the combination of 3% orange EO and 1% lemon EO in the diet of farmed fish, improved their health status [12]. In addition, citrus flavors are common in fish seasonings, so their incorporation into fish products may be highly accepted by consumers [13,14].

The genus *Cinnamomum*, belonging to the family *Lauraceae*, has different species with quite different leaf EO compositions and properties. Regarding *C. camphora*, there are several chemotypes depending on the most abundant compound in the EO: cineole-type, borneol-type, linalool-type, camphor-type, and isonerolidol-type [15]. In addition to its antibacterial properties, *C. camphora* rich in linalool has proven to have promising applications for its inclusion in nanocellulose emulsions, such that it can be incorporated into starch-based films for possible applications in food packaging. This EO strongly interacts with starch via hydrogen bonds, providing films with excellent mechanical properties [16], whereas other essential oils weaken the structure of starch. The antibacterial mechanism of linalool against *Escherichia coli* has been explained by electron donation activating processes that cause cell membrane rupture, followed by increased permeability and the loss of intracellular material [17]. Other *C. camphora* with a lower percentage of linalool (26.6%) have also proven high antibacterial activities against several bacteria, including methicillin-resistant *Staphylococcus aureus* through the same mechanism [18]. This essential oil (57.57% linalool) has also been proven to inhibit the biofilm formation and mobility of several bacteria [19]. *C. camphora* of a linalool chemotype has proven insecticidal activity due to the presence of linalool [20].

Mancuso et al. [21] evaluated the antimicrobial and antifungal properties of different EOs obtained from citrus fruits. They reported that lemon was the fruit richest in EO content, and that it was more effective against Gram-positive than Gram-negative bacteria, probably due to differences in the structure of the cell wall. However, among Gram-negative bacteria, the most sensitive to citrus EOs were those isolated from fish. Lemon EO showed a high antibacterial activity but low antifungal activity. The main components of the tested lemon EO were α-limonene (68.46%), followed by β-pinene (14.92), γ-terpinene (7.82), sabinene (2.6), α-pinene (2.45), and myrcene (2.16), explaining over 98% of the total composition. Another relevant finding was that lemon EO had higher antibacterial activity than isolated limonene. Citrus EOs and lemon EO in particular have been proven to have a high potential to be included in active packaging materials to extend the shelf life of food.

Hussain et al. [2] recently reviewed the use of EOs for the preservation of fish products. The incorporation of EOs into fish fillets, and food in general, is mainly included in hurdle technology strategies for preservation. Different systems of applying EOs to fish have been tested: (i) directly applied to the food, such as application by dipping or direct application, (ii) as part of active packaging, such as application to a vapor phase [22], or (iii) film wrapping, included in edible coatings (chitosan, starch, gelatin, mixtures, and others) at different degrees of emulsification (nanoemulsions). The latter two allow for the gradual release of active compounds and are more effective than direct application [4]. Refrigerated or superchilled storage together with vacuum or modified atmosphere packaging are the most common technologies combined with EO for fish preservation.

The aim of the present study was to evaluate the microbiological quality of vacuum-packed meat of rainbow trout (*Oncorhynchus mykiss*) treated with directly applied 0.5% and 1% lemon essential oil and 0.5% and 1% *C. camphora* essential oil.

## 2. Materials and Methods

Chemical composition of essential oils: Chemical characterization of essential oil samples was determined by gas chromatography/mass spectrometry (GC/MS) and gas chromatography (GC-FID), according to method provided by Kluga et al. [8]. 

The individual volatile constituents of injected essential oil samples were identified based on their experimentally determined retention indices and by comparison with reference spectra (WileyNIST 05 databases). The percentages of the identified compounds (amounts higher than 0.1%) were derived from their GC peak areas. 

Fish handling and packaging: The fish were caught in a reservoir in Hriňová, in central Slovakia. The fish were slaughtered and then transported to the laboratory at a temperature of 4 °C. Meat of rainbow trout (*Oncorhynchus mykiss) (Hriňová*, *Slovakia)* was cut into small pieces weighing 5 g under sterile conditions. Selected samples were treated with 0.5% and 1% lemon essential oil (Hanus, Nitra, Slovakia) and 0.5% and 1% *C. camphora* essential oil (Hanus, Nitra, Slovakia). Vacuum-packed samples were packed using a vacuum packaging machine (Concept, Choceň, Czech Republic). A total of 105 samples were prepared. Seven types of samples were prepared: (1) control group with air—fish packed into polyethylene bags and stored at 4 °C; (2) control group with vacuum packaging—vacuum-packed fish, stored at 4 °C; (3) control group with sunflower oil—vacuum-packed fish with sunflower oil, stored at 4 °C; (4) vacuum-packed fish treated with 0.5% lemon essential oil, stored at 4 °C; (5) vacuum-packed fish treated with 1% lemon essential oil, stored at 4 °C; (6) vacuum-packed fish treated with 0.5% *C. camphora* essential oil, stored at 4 °C; and (7) vacuum-packed fish treated with 1% *C. camphora* essential oil, stored at 4 °C.

Sampling and cultivation: A total of 105 samples were analyzed. Microbiological analyses were performed on the 0th, 1st, 3rd, 5th, and 7th day of storage. Samples of rainbow trout, weighing 5 g, were placed in Erlenmeyer flasks and 45 mL of 0.1% sterile saline solution were added to each Erlenmeyer flask. The prepared samples were placed in a shaker and homogenized. Plate Count Agar (PCA, Oxoid, Basingstoke, UK) was used for total viable counts (TVCs). Inoculated Petri dishes were incubated at 30 °C for 72 h. Violet Red Bile Lactose Agar (VRBL, Oxoid, Basingstoke, UK) was used for the cultivation of coliform bacteria (CB). Inoculated Petri dishes were incubated at 37 °C for 48 h. De Man, Rogosa and Sharpe agar (MRS, Oxoid, Basingstoke, UK) was used for the cultivation of lactic acid bacteria (LAB). Incubation was performed in an incubator under an atmosphere containing 5% CO_2_ at 37 °C for 48–72 h. 

Identification of microorganisms with a MALDI-TOF MS Biotyper: A MALDI-TOF MS Biotyper mass spectrometer (Bruker, Daltonics, Bremen, Germany) was used to identify isolated microorganisms from samples. MALDI-TOF MS (matrix-assisted laser desorption/ionization–time-of-flight mass spectrometry) has the advantage of speed, low cost, and simplicity, and emerges as an attractive option for microbial typing. The analysis workflow used for typing by MALDI-TOF MS is the same as for bacterial identification, and includes the culturing and preparation of samples, protein extraction with ethanol–formic acid, spectrum acquisition, and data analysis.

Preparation of MALDI matrix solution: The stock solution consisted of 50% acetonitrile, 47.5% water, and 2.5% trifluoroacetic acid. There were pipetted 500 μL of 100% acetonitrile, 475 μL of distilled water, and 25 μL of 100% trifluoroacetic acid into an Eppendorf tube. Subsequently, 250 μL of organic solvent with ‘HCCA matrix portioned’ was added to the Eppendorf tube.

Sample preparation and identification: The colonies were resuspended in 300 μL of sterile distilled water after incubation, and 900 μL of absolute ethanol was added. The mixture was centrifuged at 10,000× *g* for 2 min. The pellet was centrifuged again after the supernatant was discarded. The precipitate was allowed to dry at room temperature. Then, 30 μL of formic acid (70%) and 30 μL of acetonitrile were added and mixed thoroughly with the pellet. The solution was centrifuged at maximum speed for 2 min, and 1.5 μL of the supernatant was spotted on a polished MALDI target plate (Bruker Daltonics, Bremen, Germany). Then, 1.5 μL of the matrix solution was added to each spot and allowed to dry. Immediately after drying, the samples were processed on a MALDI-TOF MS spectrometer with flexControl software (BrukerDaltonics, Bremen, Germany). Each spectrum was obtained by averaging 40 laser shots obtained in automatic mode with the minimum laser power necessary to ionize the samples. The spectra were analyzed and compared to the database according to real-time software, v3.1 classification [23].

Statistical analysis: All measurements and analyses were carried out in triplicate. The experimental data were evaluated by using IBM SPSS Statistics 26 (IBM Corporation, Armonk, New York, NY, USA). Analysis of variance (ANOVA) was used to evaluate the results. Correlation analyses were performed using an F-test and Tukey’s HSD test. Levene’s test was run to check the homogeneity of variances.

## 3. Results

### 3.1. Chemical Composition of Essential Oils

The chemical composition of *C. caphora* var. *Linalolifera*, used in the study, is presented in Table 1. Linalool is the major compound, accounting for 98.1% of the total profile; if other forms of linalool are considered, they make up to 99.5% of the total profile. Linalool is a terpene with an alcohol group, and is expected to be the major compound in the EO of this chemotype of *C. camphora*.

The composition of *C. limon* EO is presented in Table 2. As expected, most of the components are monoterpenes; the main compound is α-limonene (58.9%), followed by β-pinene (13.3%) and γ-terpinene (11.2%).

### 3.2. Microbiological Quality of Fish Meat

The average number of TVCs ranged from 3.07 ± 0.07 log CFU/g on day 0 to 4.49 ± 0.20 log CFU/g on the seventh day in control samples stored in aerobic conditions; from 3.03 ± 0.07 log CFU/g on the first day to 3.96 ± 0.07 log CFU/g on the seventh day in vacuum-packed samples; from 2.87 ± 0.08 log CFU/g on the first day to 3.53 ± 0.11 log CFU/g on the seventh day in vacuum-packed samples with sunflower oil; from 2.85 ± 0.12 log CFU/g on the third day to 3.15 ± 0.14 log CFU/g on the seventh day of storage in samples with 0.5% lemon EO; from 2.60 ± 0.09 log CFU/g on the third day to 3.03 ± 0.14 log CFU/g on the seventh day in samples with 1% lemon EO; from 2.67 ± 0.09 log CFU/g on the third day to 3.01 ± 0.14 log CFU/g on seventh day in samples with 0.5% *C. camphora* EO; and from 2.53 ± 0.11 log CFU/g on the third day to 2.93 ± 0.15 log CFU/g on the seventh day of storage in samples with 1% *C. camphora* EO (Figure 1). The number of TVCs in samples treated with lemon EO and in samples treated with *C. camphora* EO were significantly lower (*p* < 0.05) compared with the control samples during all days of storage, and also lower than those of the vacuum control and the sunflower control, showing a clear antimicrobial effect of the addition of both EOs. Vacuum packaging enhanced the microbial quality by 0.5 log units compared to the control; additionally, the sunflower control enhanced the microbial quality by a 1 log unit decrease.

The average number of CB ranged from 1.18 ± 0.08 log CFU/g on day 0 to 2.65 ± 0.11 log CFU/g on the seventh day in aerobically packaged samples; from 1.03 ± 0.05 log CFU/g on the first day to 2.41 ± 0.12 log CFU/g on the seventh day in vacuum-packed samples; from 1.18 ± 0.08 log CFU/g on day 0 to 2.29 ± 0.09 log CFU/g on the seventh day of storage in samples with sunflower oil; from not detected on the first and third day of storage to 2.11 ± 0.10 log CFU/g on the seventh day in samples with 0.5% lemon EO; from 1.18 ± 0.08 log CFU/g on day 0 to 2.13 ± 0.10 log CFU/g on the seventh day in samples with 0.5% *C. camphora* EO; and from 1.18 ± 0.08 log CFU/g on day 0 to 1.20 ± 0.10 log CFU/g on the seventh day of storage in samples treated with 1% *C. camphora* EO. Coliform bacteria were not present on the first, third, fifth, and seventh day of storage in samples treated with 1% lemon EO (Figure 2). Only the application of EOs with a concentration of 1% enhanced the microbial quality of fish filets at day seven (*p* < 0.05), lemon being the most effective EO. All EO concentration were effective (*p* < 0.05) at controlling CB during the first three days of storage.

LAB were not present in air-packed and in vacuum-packed fish during the seven days of storage, and not detected in lemon-EO-treated fish until the seventh day. However in *C. camphora* and sunflower-treated fish they were detected at the fifth storage day (Figure 3). Detected counts of LAB ranged from 1.05 ± 0.04 log CFU/g on the fifth day to 1.39 ± 0.04 log CFU/g on the seventh day. Very low counts of LAB have been detected in the present study, and they were only present when oil or EOs were added.

### 3.3. Identification of Isolated Microorganisms from Samples Using a MALDI-TOF MS Biotyper 

From each of the used microbial media, colonies were isolated and identified as described previously. We present isolated species under all tested treatments (Table 3) as a function of the medium of isolation, and a global classification of the isolated bacteria into families (Table 4).

*Lactobacillus sakei* was isolated from lactic acid bacteria media in three cases and *Carnobacterium maltaromaticum* in addition to *Staphylococcus hominis* in one case. *Lactobacillus sakei* was isolated from a control sample treated with sunflower oil as well as from samples treated with 0.5% and 1% *C. camphora* essential oil. *Carnobacterium maltaromaticum* was isolated from a control with sunflower oil and vacuum-packed samples, and *Staphylococcus hominis* was isolated from samples treated with 0.5% lemon EO.

The genus *Aeromonas* was isolated from samples of all treatments. *Aeromonas salmonicida*, *Aeromonas bestiarum*, and *Aeromonas eucrenophila* were isolated in seven cases, Aeromonas sobria in three cases, *Aeromonas molluscorum* and *Aeromonas popoffii* in two cases, and Aeromonas media in one case. *Pseudomonas proteolytica* was isolated in five cases, *Pseudomonas fragi* in three cases, and *Pseudomonas libanensis*, *Pseudomonas lundensis*, *Pseudomonas rhodesiae*, *Pseudomonas taetrolens*, *Pseudomonas gessardii*, and *Pseudomoas bressnerii* in one case. *Acinetobacter johnsonii*, *Acinetobacter harbinensis* and *Bacillus altitudinis* were isolated from control air-packed samples. *Brochothrix thermosphacta* was isolated from samples treated with 1% lemon EO (Table 3).

Table 4 shows the classification of individual microorganisms into families.

Most of the isolated bacteria belong to the family Pseudomonadaceae (29%); the family with the second highest proportion was Aeromonadaceae (26%). The following families were also represented: Enterobacteriaceae (18%), Moraxellaceae (7%), Staphylococcaceae (4%), Lactobacillaceae (4%), Listeriaceae (4%), Carnobacteriaceae (4%), and Bacillaceae (4%) (Figure 4).

## 4. Discussion

The composition of the EOs used in the present study showed high similarities with those of Klüga et al. [8]. Few studies are available on *C. camphora*, and specifically on the chemotype Linaloolifera. Other authors using this chemotype of *C. camphora* EO reported much lower contents of linalool, as low as 26.6% [18] and 57.6% [19], although they still reported good antimicrobial activities of the EO. The tested oil can be almost considered as pure linalool. Regarding *C. limon* EO, the composition is quite similar to that reported by other authors [8,14], with its main component being α-limonene. In the scientific literature it has been reported that the major component of lemon EO is α-limonene, in the range of 51% [24] to 68% [21], together with other monoterpenes to complete the profile. Lemon EO is one of the main EOs in the market. It is obtained from citrus juice co-products, and it is available in high quantities for different purposes (mainly food, cosmetics, and pharmaceuticals), which may be the reason why the composition of commercial lemon EOs has a lower variability than those of *C. camphora*.

Lemon EO at a concentration of 1% has proven a good control of TVCs, and even better control of CB and LAB in this study. Other authors tested lemon EO to preserve other fish, such as salted sardines [14], reporting an EO composition of α-limonene (59.84%), β-pinene (12.74), *p*-cymene (7.23), and γ-terpinene (3.78), and contents of sabinene, α-pinene, β-myrcene, 4-terpineol, α-terpineol, β-citral, α-citral, and neryl acetate between 1–2%. The addition of lemon EO reduced the formation of histamine, affected microbial populations, and had positive effects on the sensory quality of salted sardines. The major component of lemon EO, limonene, has been tested for the preservation of fresh fish (gilthead sea bream fillets). Limonene mixed with sunflower oil (0.8% to 1.6% EO in the oil) was applied over the fillets followed by vacuum packaging and refrigerated storage at 2 °C [13]. Specific spoilage bacteria isolated from fish (from genus *Pseudomonas* and *Shewanella*) were evaluated during 15 days of refrigerated storage and sensory analysis of fish was carried out. The addition of limonene improved fish microbial quality and sensory scores in a concentration-dependent manner. These results are similar to those in the present study where lemon EO has proven good inhibition of CB; species of the genus *Pseudomonas* were isolated from fish samples under all treatments except those of the vegetable oil control. Coliform bacteria and Enterobacteriaceae are hygienic indicators. They indicate contamination and the possible presence of pathogenic bacteria in fish and aquatic environments [25]. Lemon EOs have shown good properties as inhibitors of CB.

Several other citrus EOs have been applied to sardines to evaluate their preservation [24]. Lemon EO proved the best antimicrobial activity against *Staphylococcus aureus* in in vitro assays, whereas bergamot was the most effective in real food samples. Limonene was the most abundant compound in lemon EO: α-limonene (51.39%), β-pinene (17.04%), and γ-terpinene (13.46%); in bergamot linalyl acetate and linalool were the most abundant. In the present study, only one species of *Staphylococcus* was isolated. As it was from 0.5%-lemon-EO-treated fish, so we cannot really compare the effectiveness of lemon EO on *S. aureus*.

Regarding previous studies in the preservation of rainbow trout by using EOs, the use of *C. camphora* var. Linaloolifera has not been reported, whereas there are a few studies reporting the use of other types of cinnamon and citrus oils. In the present paper, we use the term cinnamon to indicate that the EO belongs to species *C. verum*, *C. cassia*, or *C. ceylanicum*, where the main component is cinamaldehyde. Cinnamon essential oil proved strong antimicrobial activity in food, including fishery products [26]. Cinnamon and other EOs (rosemary, fennel, and cardamom) were added to carp fingers to evaluate their effect on microbial counts (total bacteria, psychrophilic bacteria, molds, and yeasts), target species *(Bacillus cereus*, *Staphylococcus aureus*, and *Escherichia coli*), and quality indicators (oxidation, proteolysis, and sensory analysis) [27]. All tested oils when applied extended carp shelf life, rosemary being the most effective, followed by cinnamon. In the present study, both tested oils significantly reduced counts of TVCs. Zhang et al. [28] reported that the treatment of carp with cinnamon oil in combination with vacuum packaging prolonged its shelf life by 2 days compared to untreated samples. The number of TVCs exceeded 10 log CFU/g in untreated samples on the tenth day, and the number of TVCs was 7.1 log CFU/g in samples treated with cinnamon essential oil on the twelfth day of storage. No such high counts were detected in the present study under any of the studied packaging conditions; initial counts (about 3 log CFU/g TVC and 1 log CFU/g EB) show a good microbial quality of the studied fish. Van Haute et al. [29] reported that cinnamon EO had no antimicrobial effect on lactic acid bacteria in salmon meat stored under vacuum and in a modified atmosphere. Although a different type of cinnamon and fish have been tested in the present study, the presence of *C. camphora* EO allowed the growth of lactic acid bacteria. Gelatin coatings including cinnamon EO (from 1% to 2%) were successfully applied on rainbow trout fillets, and their shelf life was extended (reduced bacterial counts and quality preservation) up to 15 days of refrigerated storage [30].

Regarding studies on lemon EO for the preservation of rainbow trout: Nanoemulsions of citrus EOs, including lemon, have been tested on rainbow trout fillets [31]. All citrus EOs extended the shelf life of the fillets (as assessed by microbial counts and biochemical parameters), the most effective being mandarin and grapefruit EOs (16 days shelf life) followed by orange and lemon (14 days), whereas the control had 10 days of shelf life. The inhibitory effect of EOs used in our study was limited after 7 days. Lemon extracts (aqueous) have also been tested for the preservation of fish products and proved to have synergistic effects with other extracts (grapeseed and thymol) for the extension of the shelf life of fish hamburgers (reduced microbial counts and delay on sensory decay) [32]. Essential oil of lime (*Citrus latifolia*) included in a chitosan–gelatin coating was applied to rainbow trout fillets further stored at superchilled conditions [33]. As a result, the microbial quality of the fillets was enhanced (delayed growth of total mesophilic and psychrotrophic bacteria, enterobacteria, and lactic acid bacteria). Those results agree with those observed in the present study.

Many other EOs have been tested to extend the shelf life of rainbow trout. Some of studies point out the fact that higher EO concentrations may greatly impair the sensory properties of the product. An EO reported to be effective in the control of *Listeria monocytogenes* in rainbow trout fillets was *Zataria multiflora* (66% carvacrol, 26% thymol, and 4% *p*-cymene) at concentrations of 0.8 and 1.5% in films [34]. The inclusion of *Bunium persicum* EO at 2% [35] and *Foenniculum vulgare* at 2% [36] in nanocoatings was effective in reducing the microbial spoilage of rainbow trout fillets. *Coriandrum sativum* EO at a 0.5% addition to rainbow trout fillets increased the shelf life and improved the microbial and sensory quality of the fillets [37]. *Ferulago angulata* EO (22% cis-β-O Cymene and 22% α-pinene) was also tested, 3% nanoemulsified in chitosan film, for the preservation of refrigerated rainbow trout fillets, successfully improving antioxidant and antimicrobial properties (against *Shewanella putrefaciens* and *Pseudomonas fluorescens*) as well as improving sensory properties [38]. Lemon verbena EO 0.5% and extracts 1% included in chitosan applied as an edible coating to rainbow trout fillets followed by vacuum packaging significantly improved microbial quality (reduced counts of TVCs, psychrotrophic bacteria, Enterobacteria, and H_2_S bacteria), delayed fish spoilage, and enhanced sensory properties [39]. Kuzgun [40] investigated the effect of different concentrations of garlic essential oil (1%, 2% and 4%) on the microbiological quality of rainbow trout fillets for 15 days stored at 2 °C. The number of LAB, TVCs, and yeast was significantly higher in the control sample (*p* < 0.05) compared to the treated samples. The lowest number of LAB was recorded in a sample treated with 4% garlic essential oil on the 15th day. Aksoy and Sezer [41] discovered that the combination of vacuum packaging and 2% laurel essential oil extended the shelf life of rainbow trout filets by approximately 4 days. Çoban et al. [42] found that the combination of the vacuum packing and 2% sage essential oil resulted in an extension of the microbiological shelf life by 24 days and the combination of the vacuum packing and 4% sage essential oil by 34 days. In our study, the inhibitory effect of EOs against LAB in fish was limited to the first 5 days of storage; other EOs have different inhibitory activity against LAB. Angiş and Oğuzhan [43] compared the combined effect of thyme essential oil and packaging (vacuum and modified atmosphere) on fresh rainbow trout fillets during 18 days of storage at 4 °C. The bacterial growth of lactic acid bacteria (LAB) was inhibited in all samples treated with thyme essential oil, but the combination with the modified atmosphere showed a better inhibitory effect than the combination with vacuum packaging. However, different results were obtained by Yildiz [44], who investigated the effect of 1% thyme essential oil and 1% rosemary essential on the quality of marinated rainbow trout during storage at 4 °C. The initial number of LAB was 2.0 log CFU/g. The number of LAB was 6.47 log CFU/g in samples treated with rosemary EO and a 5.54 log CFU/g in samples treated with thyme EO at the end of storage. Caraway and rosemary EOs could be effective in inhibiting the pathogenic *Lactococcus garvieae* in rainbow trout [45], as could mint EO [46].

In general, the reviewed scientific literature points to an enhanced preservation effect when EOs are applied combined with vacuum packaging, as reported in the present study for the application of lemon and *C. camphora* EOs. TVCs, EB, and LAB are the main reported microbial indicators analyzed in the literature, as in the present study, and the most effective EOs are those containing carvacrol and thymol (rosemary, zataria, etc.). The concentration of EO necessary to reach effectiveness may be high (as much as 4% for garlic and sage), whereas most studies tend to test concentrations under 2% EO. In the present study, successful results have been obtained for EO concentrations of 1%. It has to be considered that when high concentrations of EOs are used, changes in organoleptic properties are dramatic, so the potential culinary uses are reduced to few preparations. Usually EO concentration are kept below 3% to avoid off-flavors [4], and sometimes even lower, between 0.5–1% [1]. EOs of lemongrass, garlic, oregano, rosemary, lemon, lime, onion, and thyme, applied at a 0.9%, were effective in reducing counts of Listeria in salmon, but the strong flavors and odors yielded low sensory scores [47]. Consequently, it is necessary to develop strategies of hurdle technology to keep EO concentrations as low as possible by combining their use with other technologies (superchilling, vacuum or modified atmosphere packaging, and emerging technologies), which have been proven to have synergistic effects in most cases [48]. The aroma of linalool has been described as floral, citric, fruity, and fresh [49], so it is expected to be milder than that of other EOs such as garlic or zataria. Lemon EO has a clear citric aroma mainly due to limonene, and although both tested EOs provide aromas to fish, they are mild and quite common in fish seasonings.

The identification of the microflora of freshwater fish is an important tool for assessing the quality and safety of fish intended for human consumption [25]. Klūga et al. [25] isolated the genera *Pseudomonas* spp. (55%), *Pantoea* spp. (9%), *Serratia* spp. (7%), and *Rahnella* spp. (7%) from 23 freshwater fish. Jalal et al. [50] compared the presence of pathogenic microorganisms and microorganisms causing the spoilage of freshwater and marine fish. They found that the dominant microorganisms of freshwater fish are *Vibrio* spp., *Enterobacter* spp., *Serratia* spp., and *Aeromonas* spp. Sørensen et al. [51] isolated the genera *Pseudomonas*, *Photobacterium*, *Shewanella*, and *Acinetobacter* from fresh chilled cod. Yagoub [52] isolated the Enteriobacteriaceae family and *Pseudomonas* spp. from fresh fish bought in the supermarket (*Tilapia nilotica* Linn). Among the bacteria from the Enterobacteriaceae family, 23.2% of the strains isolated were identified as *E. coli*. In the studied rainbow trout, most isolated families were Pseudomonadaceae and Aeromonadaceae. The knowledge of fish microbiota and its modification as an effect of EO addition may help in developing strategies to enhance fish preservation in search of synergistic compounds of hurdle technology strategies to enhance fish quality and safety.

## 5. Conclusions

*C. camphora* (98% linalool) and *C. lemon* (58% α-limonene) have a positive effect on the microbial quality of fish meat when added to rainbow trout filets. *C. camphora* EO at a concentration of 1% in rainbow trout and combined with vacuum packaging is highly effective against the bacterial growth of total viable counts and lactic acid bacteria. *C. lemon* EO at a concentration of 1% is the most effective against coliform bacteria. The use of a combination of essential oils with vacuum packaging has the benefit of extending the shelf life of rainbow trout meat, and it would therefore be appropriate to use them in practice. Dominant microbial families isolated from rainbow trout fillets are Pseudomonadaceae and Aeromonadae.

## Figures and Tables

**Figure 1 animals-11-03145-f001:**
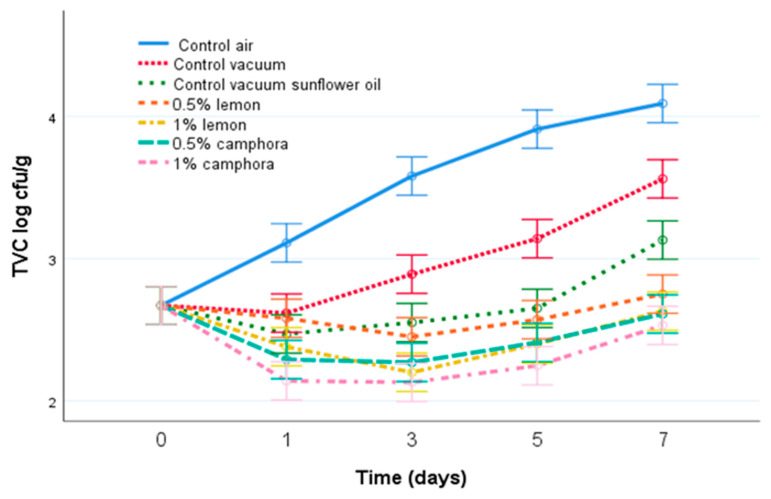
Average numbers (log cfu/g) of TVCs in samples of rainbow trout during 7 days of storage at 4 °C. Air control—aerobically packed control samples; vacuum control—vacuum-packed control samples; vegetable oil control—vacuum-packed control samples treated with sunflower oil; lemon EO 0.5—vacuum-packed samples treated with 0.5% lemon EO; lemon EO 1—vacuum-packed samples treated with 1.0% lemon EO; *C. camphora* EO 0.5—vacuum-packed samples treated with 0.5% *C. camphora* EO; and *C. camphora* EO 1—vacuum-packed samples treated with 1.0% *C. camphora* EO. Bars denote ± twice the standard error (*p* < 0.05).

**Figure 2 animals-11-03145-f002:**
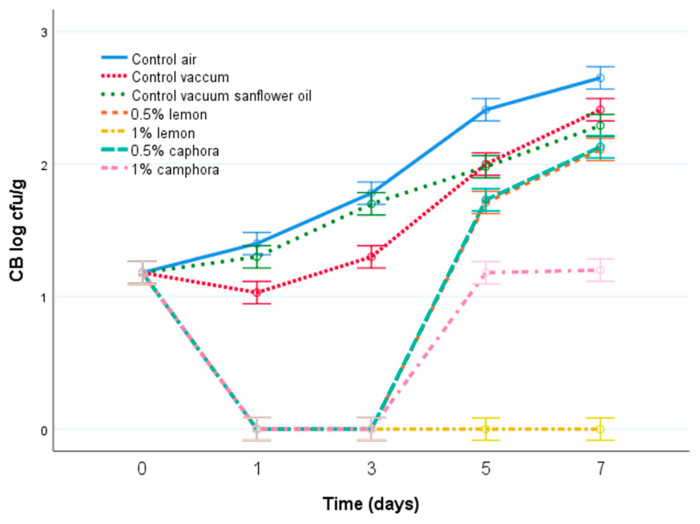
Evolution of counts (log cfu/g) of CB in samples of rainbow trout during 7 days of storage at 4 °C. Air control—aerobically packed control samples; vacuum control—vacuum-packed control samples; vegetable oil control—vacuum-packed control samples treated with sunflower oil; lemon EO 0.5—vacuum-packed samples treated with 0.5% lemon EO; lemon EO 1—vacuum-packed samples treated with 1.0% lemon EO; *C. camphora* EO 0.5—vacuum-packed samples treated with 0.5% *C. camphora* EO; and *C. camphora* EO 1—vacuum-packed samples treated with 1.0% *C. camphora* EO. Bars denote ± twice the standard error (*p* < 0.05).

**Figure 3 animals-11-03145-f003:**
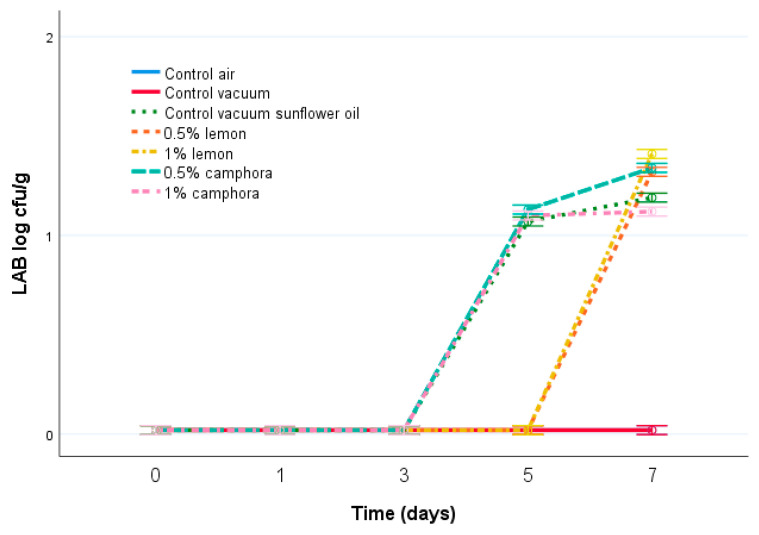
Evolution of counts of LAB (log cfu/g) in samples of rainbow trout during 7 days of storage at 4 °C. Air control—aerobically packed control samples; vacuum control—vacuum-packed control samples; vegetable oil control—vacuum-packed control samples treated with sunflower oil; lemon EO 0.5—vacuum-packed samples treated with 0.5% lemon EO; lemon EO 1—vacuum-packed samples treated with 1.0% lemon EO; *C. camphora* EO 0.5—vacuum-packed samples treated with 0.5% *C. camphora* EO; and *C. camphora* EO 1—vacuum-packed samples treated with 1.0% *C. camphora* EO. Bars denote ± twice the standard error (*p* < 0.05).

**Figure 4 animals-11-03145-f004:**
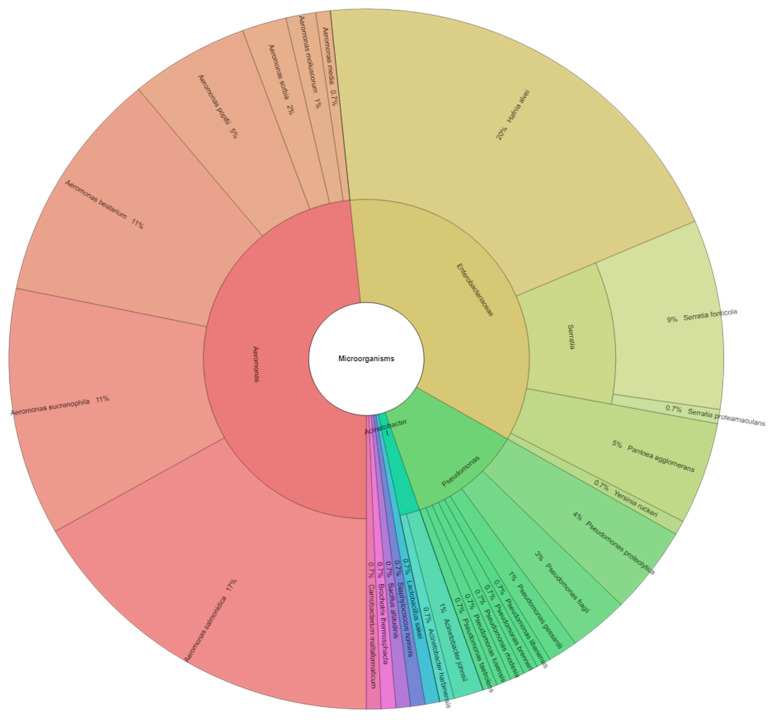
Krona chart of percentage representation of bacterial families.

**Table 1 animals-11-03145-t001:** Chemical composition of essential oil from *Cinnamomum caphora* var. *Linalolifera*.

No	RI ^a^	Compound ^b^	% ^c^
1	992	β-myrcene	tr
2	1004	α-phellandrene	tr
3	1016	α-terpinene	tr
4	1023	*p*-cimene	tr
5	1028	α-limonene	tr
6	1029	β-phellandrene	tr
7	1033	1,8-cineole	tr
8	1034	Benzyl alcohol	tr
9	1038	(*Z*)-β-ocimene	tr
10	1047	(*E*)-β-ocimene	tr
11	1060	γ-terpinene	tr
12	1074	Cis-linalool oxide	0.6
13	1089	Trans-linalool oxide	0.9
14	1098	linalool	98.1
15	1148	camphor	tr
16	1422	(*E*)-caryophyllene	tr
	Total		99.5

^a^ Values of retention indices on HP-5MS column; ^b^ identified compounds; and ^c^ tr—compounds identified in amounts less than 0.1%, RI—retention index.

**Table 2 animals-11-03145-t002:** Chemical composition of essential oil from *Citrus limon*.

No	RI ^a^	Compound ^b^	% ^c^
1	926	α-thujene	0.6
2	938	α-pinene	2.6
3	948	Camphene	tr
4	977	Sabinene	2.8
5	980	β-pinene	13.3
6	992	β-myrcene	2.0
7	998	n-octanal	tr
8	1004	α-phellandrene	tr
9	1016	α-terpinene	0.2
10	1023	*p*-cimene	1.0
11	1028	α-limonene	58.9
12	1047	(*E*)-β-ocimene	tr
13	1060	γ-terpinene	11.2
14	1088	α-terpinolene	0.5
15	1103	n-nonanal	tr
16	1136	Cis-limonene oxide	tr
17	1138	Trans-limonene oxide	tr
18	1152	Citronellal	tr
19	1178	4-terpinenol	tr
20	1189	α-terpineol	0.2
21	1202	n-decanal	tr
22	1238	Neral	1.1
26	1266	Geranial	1.8
27	1355	Citronellyl acetate	tr
28	1364	Neryl acetate	0.6
29	1380	Geranyl acetate	0.3
30	1422	(*E*)-caryophyllene	0.3
31	1437	α-trans-bergamotene	0.6
32	1497	Valencene	tr
33	1507	β-bisabolene	0.9
	total		98.8

^a^ Values of retention indices on HP-5MS column; ^b^ identified compounds; and ^c^ tr—compounds identified in amounts less than 0.1%, RI—retention index.

**Table 3 animals-11-03145-t003:** Isolated bacteria from samples of rainbow trout.

Sample	CB	LAB	Other
Control—air	*Hafnia alvei*, *Serratia fonticola*, *Serratia proteamaculans*, *Yersinia ruckeri*, and *Pantoea agglomerans*		*Bacillus altitudinis*, *Aeromonas salmonicida*, *Pseudomonas libanensis*, *Pseudomonas proteolytica*, *Pseudomonas fragi*, *Acinetobacter johnsonii*, and *Acinetobacter harbinensis*
Control—vacuum	*Hafnia alvei*		*Aeromonas bestiarum*, *Aeromonas salmonicida*, *Aeromonas eucrenophila*, *Pseudomonas fragi*, and *Pseudomonas proteolytica*
Control—sunflower oil	*Hafnia alvei*, *Serratia fonticola*, and *Pantoea agglomerans*	*Lactobacillus sakei* *Carnobacterium maltaromaticum*	*Aeromonas eucrenophila*, *Aeromonas bestiarum*, *Aeromonas molluscorum*, *Aeromonas salmonicida*, *Aeromonas popoffii*, and *Aeromonas sobria*
Lemon EO 0.5	*Hafnia alvei* and *Serratia fonticola*		*Aeromonas eucrenophila*, *Aeromonas popoffii*, *Aeromonas salmonicida*, *Aeromonas bestiarum*, *Aeromonas media*, *Aeromonas sobria*, *Pseudomonas lundensis*, *Pseudomonas proteolytica*, and *Staphylococcus hominis*
Lemon EO 1	*Hafnia alvei* and *Serratia fonticola*		*Aeromonas bestiarum*, *Aeromonas eucrenophila*, *Aeromonas salmonicida*, *Aeromonas sobria*, *Pseudomonas rhodesiae*, *Pseudomonas proteolytica*, and *Brochothrix thermosphacta*
*C. camphora* EO 0.5	*Hafnia alvei* and *Serratia fonticola*	*Lactobacillus sakei*	*Aeromonas bestiarum*, *Aeromonas salmonicida*, *Aeromonas eucrenophila*, *Aeromonas molluscorum*, *Pseudomonas brenneri*, and *Pseudomonas taetrolens*
*C. camphora* EO 1	*Hafnia alvei*	*Lactobacillus sakei*	*Aeromonas eucrenophila*, *Aeromonas salmonicida*, *Aeromonas bestiarum*, *Pseudomonas fragi*, *Pseudomonas proteolytica*, and *Pseudomonas gessardii*

CB—coliform bacteria, LAB—lactic acid bacteria.

**Table 4 animals-11-03145-t004:** Families of isolated microorganisms.

Microorganisms	Family
*Lactobacillus sakei*	Lactobacillaceace
*Staphylococcus hominis*	Staphylococcaceae
*Pseudomonas gessardii*, *Pseudomonas libanensis*, *Pseudomonas fragii*, *Pseudomonas proteolytica*, *Pseudomonas brenneri*, *Pseudomonas rhodesia*, *Pseudomonas lundensis*, and *Pseudomonas taetrolens*	Pseudomonadaceae
*Hafnia alvei*, *Serratia fonticola*, *Serratia proteamaculans*, *Yersinia ruckeri*, and *Pantoea agglomerans*	Enterobacteriaceae
*Bacillus altitudinis*	Bacilliaceae
*Aeromonas salmonicida*, *Aeromonas popoffii*, *Aeromonas eucrenophila*, *Aeromonas bestiarum*, *Aeromonas molluscorum*, *Aeromonas sobria*, and *Aeromonas media*	Aeromonadaceae
*Acinetobacter johnsonii Acinetobacter harbinensis*	Moraxellaceae
*Brochothrix thermosphacta*	Listeriaceae
*Carnobacterium maltaromaticum*	Carnobacteriaceae

## Data Availability

The data presented in this study are available on request from the corresponding author.

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
