# Peer review of "Influence of Essential Oils on the Microbiological Quality of Fish Meat during Storage"

_animals, 2021, doi:10.3390/ani11113145_

Round 1
Reviewer 1 Report
Paper is highly descriptive. The authors report in the results the numbers (minimum and maximum) shown in the table. Wouldn't it be better to have graphs instead of tables?
There is no statistical analyses that support the effect of the 2 essential oils and storage time at different concentrations. I suggest to make analysis of variance (ANOVA) in order to evaluate the differences among the
groups. This will allow to give greater strength to your conclusions.
The discussion, in my opinion too long, appears to be a review. For example, lines 360 to 366 and 378 to 381. The same reference that repeat twice the EO used. I think it is too detailed and moreover, many authors have studied the effect of essential oils on the whole fish fillet, while this paper used a small sample.
Author Response
Reviewer #1
We would like to thank the reviewer for comments and suggestions on the manuscript that help us in improving the quality and clarity of the manuscript.
Paper is highly descriptive.
The manuscript has been revised to shorten most descriptive parts and enhance the discussion
Point 1: The authors report in the results the numbers (minimum and maximum) shown in the table. Wouldn't it be better to have graphs instead of tables?
Response: Thank you for suggestion, authors still think that showing tables with standard deviation and the recently introduced results of ANOVA (mean comparisons) provide more information than graphs.
Point 2: There is no statistical analyses that support the effect of the 2 essential oils and storage time at different concentrations. I suggest making analysis of variance (ANOVA) in order to evaluate the differences among the groups. This will allow to give greater strength to your conclusions.
Response: The results of ANOVA are now indicated in the tables: Significant differences between sample groups were added.
Point 3: The discussion, in my opinion too long, appears to be a review. For example, lines 360 to 366 and 378 to 381.
Response: The discussion has been shortened removing those references (11 references from the indicated lines) that were not directly related and now is more focused on comparison to the results
Point 4: The same reference that repeat twice the EO used.
Response: This whole section has been revised to avoid repetition.
Point 5: I think it is too detailed and moreover, many authors have studied the effect of essential oils on the whole fish fillet, while this paper used a small sample.
Response: Although many authors have studied the influence of other combinations of essential oils on fish fillets this is the first study reporting the influence of EO from Citrus limon and Cinnamomum camphora on the quality of rainbow trout followed by identification of bacterial species by MALDI-TOF MS Biotyper. We used 5 g samples, allowing a good repeatability of conditions, we run 3 independent replicates whit a total of 105 samples analyzed, which is a good number of samples, and we do consider that the present study provides new and highly valuable data.
Thank you again for your valuable comments.
Reviewer 2 Report
the language needs to be improved. The material and methods section is not clear why only store for 7 days? please explain the use of maldi-tof biotyper better.
statements like in l 67/68 on the high potential health risk of synthetic preservatives and their toxicological effects - do the authors think the the legal food additives make the food toxic?
The discussion is a list of a lot of literature findings and this really needs to be discussed related to the authors findings. also the conclusions needs to be about the findings of the authors so the first three lines are not relevant!
Author Response
Reviewer #2
We would like to thank the reviewer for comments and suggestions on the manuscript that help us in improving the quality and clarity of the manuscript.
Point 1: The language needs to be improved.
Response: Language of manuscript has been revised and some part rewritten (see corrections in blue color).
Point 1: The material and methods section is not clear why only store for 7 days?
Response: Based on previous experiments with fish meat, we found that the fish meat is unsuitable to storage for more than 7 days.
Point 2: Please explain the use of Maldi-tof biotyper better.
Response: The MALDI-TOF MS Biotyper principle has been briefly explained in Material and methods part. Sample preparation of microorganisms and identification were deleted due to similarity report, and adequate references was added. See lines 183-195.
Point 3: statements like in l 67/68 on the high potential health risk of synthetic preservatives and their toxicological effects - do the authors think the legal food additives make the food toxic?
Response: We don’t think that legal food additives in regulated concentrations are toxic, in order to clarify the meaning the sentence has been re-written, see lines 67-68.
Point 4: The discussion is a list of a lot of literature findings, and this really needs to be discussed related to the authors findings.
Response: The discussion has been shortened removing those references (11 references) that were not directly related and now is more focused on comparison to the results
Point 5: also, the conclusions needs to be about the findings of the authors so the first three lines are not relevant!
Response: Some sentences have been deleted and conclusions revised. See lines 467-475.
Thank you again for your valuable comments.
Round 2
Reviewer 2 Report
The manuscript has been improved and some of the issues raised has been addressed.
if fish like trout and salmon only had a shelf life of 7 days it would be virtually impossible for Norway to export salmon and trout (line 378 f) please revise statement
language l 21 in healthy
l 44 and were never present
l 45 on day 7 (not seventh)
l 240 and several places
.. differ significantly
l 308 is it the oil or the results that differ?
l 322 revise language
page 12 please shorten
l 388 revise language
l 471 against bacterial growth measured as total.. as well as against growth of..
Author Response
Reviewer #1
Point 1: if fish like trout and salmon only had a shelf life of 7 days it would be virtually impossible for Norway to export salmon and trout (line 378 f) please revise statement
Response: It was revised.
language l 21 in healthy
Response: It was revised.
l 44 and were never present
Response: It was revised.
l 45 on day 7 (not seventh)
Response: It was revised.
l 240 and several places
.. differ significantly
Response: It was revised.
l 308 is it the oil or the results that differ?
Response: It was revised.
l 322 revise language
Response: It was revised.
page 12 please shorten
Response: It was revised.
l 388 revise language
Response: It was revised.
l 471 against bacterial growth measured as total.. as well as against growth of
Response: It was revised.